# Technology Readiness Drives Digital Adoption in Dentistry: Insights from a Cross-Sectional Study

**DOI:** 10.3390/healthcare13101155

**Published:** 2025-05-15

**Authors:** Christian Schnitzler, Sabine Bohnet-Joschko

**Affiliations:** Chair of Management and Innovation in Health Care, Witten/Herdecke University, Alfred-Herrhausen-Straße 50, 58455 Witten, Germany; sabine.bohnet-joschko@uni-wh.de

**Keywords:** technology readiness, digital adoption, dental clinics, digital transformation, digital equipment

## Abstract

**Background/Objectives:** Digital transformation is reshaping dentistry by improving clinical efficiency, diagnostic accuracy, and patient care. However, the adoption of digital technologies in dental clinics varies widely, influenced by multiple factors, including technology readiness. This study aimed to assess the relationship between technology readiness and digital technology adoption among German dentists, focusing on the impact of clinic characteristics and professional development. **Methods:** A cross-sectional survey was conducted among 200 licensed German dentists. Technology readiness was measured using the validated Technology Readiness Index (TRI 2.0), encompassing four dimensions: optimism, innovativeness, discomfort, and insecurity. Data on the current use of digital technologies were collected, including digital radiography, CAD/CAM systems, AI-supported tools, and patient management solutions. Statistical analyses included correlation and quartile-based comparisons to identify patterns and significant associations. **Results:** Clinics with higher TRI scores demonstrated significantly greater adoption of digital technologies. Larger clinics (MVZs) showed higher levels of digital integration compared to solo practices. Younger dentists and those engaged in continuous professional development exhibited higher technology readiness and usage of advanced digital tools. No significant gender-based differences were identified in technology readiness or digital adoption. While basic technologies like digital radiography and CAD/CAM systems were widely used, AI-based diagnostics and 3D printing remained underutilized. Key barriers included financial constraints and limited training opportunities. **Conclusions:** Technology readiness plays a critical role in shaping digital adoption in dental clinics. The findings highlight the need for targeted support, especially for smaller clinics, through professional training and investment in digital infrastructure. This study contributes to a better understanding of digital transformation in dentistry and supports strategies aligned with global health goals to improve access to digital care.

## 1. Introduction

The digital transformation in dentistry is fundamentally changing clinical workflows, improving efficiency, accuracy, and patient care [1,2,3]. Technologies such as CAD/CAM systems, artificial intelligence (AI), digital radiography, and electronic health records are becoming integral to modern dental clinics [4,5,6,7]. Despite these benefits, the adoption of digital technologies among dental practitioners varies widely, influenced by several factors, including technology readiness [8,9]. Understanding these disparities is essential to facilitating a smooth transition into digital dentistry. Since technology readiness influences how clinicians perceive and integrate new tools [10,11], assessing these factors can help ensure that all clinics, regardless of size or location, can benefit from technological advancements.

In the current landscape of dental clinics, a mixed level of digital technology adoption can be seen. While some professionals are fully equipped with the latest digital tools, others still rely on traditional methods [12,13]. This disparity raises important questions about the underlying factors influencing technology adoption. Previous research has highlighted the benefits of digital technologies in improving patient care and operational efficiency [1,2,3,4,5,6]; however, there remains a gap in understanding how technology readiness impacts the adoption of these technologies in dental clinics. Recent systematic reviews have further emphasized that, although research in robotics and artificial intelligence in dentistry is expanding, much of it remains at an early developmental stage, with limited clinical validation, highlighting a considerable gap between technological innovation and its actual clinical implementation [14].

As part of a larger research project on digital transformation in dentistry, this study provides valuable content. A previous study analyzed five key digital trends, including AI, 3D printing, and teledentistry, and projected their economic and clinical impact by 2030 [15]. While that study focused on long-term technological developments, the present research investigates technology readiness and its influence on the current adoption of digital equipment in dental clinics. By identifying key drivers and barriers, this study provides a comprehensive assessment of digital integration in dentistry today.

The Technology Readiness Index (TRI 2.0), developed by Parasuraman and Colby, is widely used to assess individual attitudes toward technology adoption [16,17]. It consists of four dimensions:The first dimension is Optimism. This dimension describes the belief in technology’s ability to enhance efficiency and control.The second dimension is Innovativeness. It is a proactive approach to adopting new technologies.The third dimension is Discomfort. It describes challenges or hesitations in using digital tools.The fourth dimension is Insecurity, meaning skepticism regarding the reliability and security of digital systems [16,17].

Previous studies have applied the TRI in various settings, highlighting its relevance in understanding technology adoption behaviors [18,19]. However, studies applying the TRI specifically within dental contexts are scarce. Research has generally focused on broader settings, revealing that younger professionals tend to display higher technology readiness and that organizational factors, such as size and type, can significantly affect digital adoption [20,21].

Germany, as Europe’s largest healthcare market, presents an important case for studying digital technology adoption in dentistry [22]. While other European countries such as Finland have successfully implemented nationwide digital health solutions [23], Germany has faced regulatory and infrastructural challenges in its digital transformation [24]. Compared to general medicine, dental clinics have been slower in adopting digital workflows, with many still relying on traditional methods [25,26].

To address these gaps, this study applies the Technology Readiness Index (TRI 2.0) to assess digital adoption in German dental clinics, evaluating key determinants such as age, clinic type, and current usage of digital equipment. Understanding how these factors shape digital transformation provides critical insights for dentists, clinic managers, and policymakers, enabling the development of targeted strategies to facilitate implementation and optimize clinical workflows.

By identifying the key drivers and barriers to digital adoption, this research offers actionable recommendations for overcoming infrastructure, financial, and training-related challenges in dental clinics. The findings align with the WHO’s and FDI’s 2030 digital health goals, which emphasize the promotion of digital solutions, reduction of healthcare disparities, and integration of innovative technologies into clinical workflows [11,27,28,29]. Strengthening technology readiness across dental practices is essential for ensuring equitable access to digital advancements and improving patient care.

## 2. Materials and Methods

This study used a cross-sectional survey design to assess the technology readiness of German dentists and its impact on the adoption of digital technologies. Germany is one of the largest healthcare markets in the world and therefore a relevant and meaningful context for analyzing technological readiness and digital acceptance in dentistry [22,30].

A power analysis was carried out analogous to our previous research [15]. The sample size for this study was determined to ensure a 95% confidence level, providing reliable representation of the target demographic [15,31]. Pilot testing indicated that approximately 85% of participants were likely to have at least one digital technology in dentistry, leading to an assumed proportion (p) of 0.85. Using the standard formula for sample size estimation, the analysis indicated that a minimum of 196 participants was required [31]. A final sample size of 200 was selected to account for potential variability and ensure robust data collection [15,32].

The survey was distributed nationwide across Germany in 2024 for a period of three months, targeting licensed dentists practicing in all federal states, with a focus on capturing diversity in clinic types and practice settings. Participants were recruited through a multi-channel distribution strategy, utilizing the reliable open-source online survey tool LimeSurvey, including professional dental associations, conferences, and online networks [15,33]. The survey was distributed nationwide across Germany to reach a broad and diverse sample of dental professionals [34]. Participation was voluntary and anonymous to limit response bias and encourage honest reporting [35]. The study was designed as a one-time survey without follow-up. Additionally, standardized definitions of digital technologies were provided within the questionnaire to reduce the risk of information bias [36].

### 2.1. Survey Instrument and Data Collection

The questionnaire was developed based on a comprehensive literature review and was divided into three main sections:Demographics—Age, gender, years of experience, clinic type (solo practice, group practice, or MVZ), and geographical location.Technology Readiness—Measured using the Technology Readiness Index (TRI 2.0), assessing the four dimensions: Optimism, Innovativeness, Discomfort, and Insecurity [17]. The TRI 2.0 captures participants’ general attitude toward the adoption of new technologies, independent of specific clinical contexts [17]. The full TRI 2.0 questionnaire has been added as Appendix A at the end of the manuscript.
To link these general attitudes to practical behavior in dental practice, additional survey Section 3 assessed the actual use of specific digital devices within operational workflows, patient management, diagnostics, and treatment planning.
3.Digital Equipment Usage—The digital technologies included in the survey were selected based on a comprehensive literature review, an analysis of current offerings from leading dental aid providers, and input from an expert panel of 15 experienced dentists. The selection focused on technologies that are currently relevant and applicable in daily clinical practice and oriented toward the patient journey (see Figure 1 in the Results section) [4,5,6,7]. Particular attention was paid to covering all critical phases of the patient journey, ensuring that the most relevant technologies were captured for each area.
Inclusion criteria for technologies were practical applicability and clinical relevance; highly experimental or purely forensic technologies were excluded.

Before distribution, the survey underwent pilot testing with 15 experienced dentists to refine clarity and relevance. Adjustments were made to enhance readability and ensure that the questions accurately captured the study objectives.

#### Independent and Dependent Variables

For the statistical analysis, dependent and independent variables recorded by the questionnaire were analyzed.

The independent variables include the following:Demographic factors such as age, gender, and years of experience.Practice characteristics like clinic size, clinic type, number of employeesProfessional development: Level of experience.The dependent variables include the following:The analysis of the Technology Readiness Score (including the TRI 2.0 total-Score and sub-dimension Scores).The number of digital technologies implemented in daily practice was also analyzed.

### 2.2. Statistical Analysis

All data were analyzed using SPSS 30. Descriptive statistics were calculated to summarize participant characteristics, TRI 2.0 scores, and digital equipment adoption rates.

To examine correlations between technology readiness and digital adoption, the following statistical analyses were conducted [37,38,39]:Pearson and Spearman correlations were tested to assess bivariate relationships.A multiple linear regression analysis was conducted to examine effects of key demographic and organizational factors on technology readiness.Quartile-based group comparisons were built to split participants into four groups according to the number of digital devices used, allowing for subgroup comparisons of technology readiness and adoption behaviors.

A *p*-value of <0.05 was considered statistically significant [40]. Confidence intervals and effect sizes were reported where applicable.

### 2.3. Participant Eligibility and Data Anonymization

Participation in this study was restricted to licensed dentists in Germany to ensure that responses reflected professional insights into digital technology adoption. Each participant confirmed their licensure status before starting the survey, maintaining data integrity.

To comply with European data protection regulations (GDPR), all responses were fully anonymized at the point of collection [41]. No personal identifiers were recorded, and demographic data were generalized to prevent traceability. These measures ensured strict confidentiality while allowing for a comprehensive analysis of technology readiness and adoption patterns.

### 2.4. Ethical Considerations

The survey focused exclusively on the integration and impact of digital technologies in dental clinics, covering aspects such as adoption rates and operational efficiency. As the study did not involve patient data, personal health information, or clinical interventions, no formal ethical review was required following consultation with the institutional ethics committee.

Participation was voluntary, with participants informed that they could withdraw at any time without consequences. The study adhered to GDPR guidelines, ensuring confidentiality, transparency, and data protection. No conflicts of interest were identified, and all procedures followed recognized ethical research standards to uphold fairness and integrity throughout the study.

## 3. Results

This section provides details on the results of the survey. First, the demographic data, technical equipment, and technology readiness are considered, and then the correlations between the individual areas are examined in detail to be able to make statements about digitalization in the German dental sector.

### 3.1. Participant Demographics

A total of 287 dentists initially participated in the survey. However, only 200 participants completed the questionnaire fully and were analyzed in this study, resulting in a dropout rate of approximately 31%. Questionnaires that were not answered in full were not taken into account. To assess potential non-response bias, the demographic characteristics of respondents and non-respondents were compared, revealing no significant differences in age, gender, or clinic type distribution.

The demographic breakdown of the participants is as follows [15]:

#### Gender Distribution

With 111 participants, 55.50% are male, and 89 participants (44.50%) are female. Table 1 shows the participants’ age distribution.

The age distribution shows that women conducted in this study are slightly younger on average than men.

These demographic data provide a comprehensive overview of the diverse backgrounds of the survey participants, highlighting a balanced representation in terms of gender and a wide age range among the respondents.

### 3.2. Current Digital Equipment in Clinics

The survey included a detailed section on the current digital equipment available in dental clinics, focusing on various aspects of digital solutions. The results provide insights into the prevalence of digital technologies across different areas of clinic operations. The digital solutions assessed in this study were identified through a comprehensive approach, including a thorough literature review, market research, and expert interviews. This multi-faceted methodology ensured that all relevant technologies currently utilized in dental practices were accurately captured and evaluated. The areas evaluated were the registration process, patient management, patient treatment, radiographic diagnostics, AI support, 3D applications and fabrication, and back-office processes [5,8,13,42,43,44,45].

Figure 1 illustrates how digital technologies are integrated across all stages of the patient journey, demonstrating that every phase is now influenced by digital components:

The data on digital equipment usage are summarized in Figure 2 and will be detailed in the following sections, following the seven sections seen in Figure 1.

I.Registration Process:

Digital solutions have made notable inroads into the registration process, with 43% of clinics already utilizing online appointment scheduling and 46% employing automated appointment reminders. However, more advanced technologies, such as AI-driven appointment management, have seen limited adoption, with only 8.5% of clinics currently incorporating these tools.

II.Patient Management:

The results clearly indicate that digital patient records have become widely established, with 85.5% of clinics adopting this technology. Additionally, the importance of online and digital anamnesis is evident, with 39.5% of clinics utilizing these tools. Automated recall systems are also in use by nearly half of all clinics (42%), reflecting their growing role in modern patient management.

III.Patient Treatment:

In the area of patient treatment, digital consent tools and paperless signatures have already been adopted by around 40% of clinics. Additionally, digital impression technology is used by every second clinic, indicating a significant level of integration into daily dental procedures.

IV.Radiographic Diagnostics and Implantation:

Fifty-six point five percent (56.5%) of clinics currently use 3D radiography, and among these users, almost all plan implants digitally. However, navigated implantology is less commonly employed, with only 33.5% of dentists utilizing this advanced technique.

V.AI-Support and Cloud storage:

Currently, AI-supported tools are used by nearly one in five dentists. Among these, 26.5% utilize AI for caries detection, 25% for radiographic analysis, and 21% for assisted dental design. Cloud storage of patient data currently has limited relevance, with only 14% of clinics using it.

VI.3D Applications:

In contrast to the limited use of AI and cloud storage, 3D printing is more widely adopted, being utilized by 32.5% of clinics, while CAD-CAM devices are even more common, with 48% of clinics integrating them into their workflows.

VII.Back-office Processes:

Paperless billing has already been adopted by one in four clinics, with 25% currently utilizing this solution. However, AI-supported tools for business analysis remain largely underutilized, with only 12.5% of clinics implementing them.

These findings highlight the varied levels of digital technology adoption across different functional areas within dental clinics. While radiographic diagnostics and the digital patient records show high levels of digital integration, AI support and cloud storage are still emerging areas with significant growth potential.

While the first analysis focused on evaluating specific types of digital equipment, showing the distribution of each per clinic, the following evaluation centers on the overall digital adoption within each clinic by assessing how many of the total tools are being utilized.

The second analysis shows that the number of digital equipment/services (equipment) per clinic varied widely, ranging from a minimum of 0 to a maximum of 20 pieces of equipment. Figure 3 shows the distribution of the equipment among the dental clinics.

For further analysis, the mean values, median values, and standard deviation were analyzed:The mean number of equipment per clinic is 7.5.The median number of equipment per clinic is 7.The standard deviation is (±) 4.6.

These statistics reveal that while the average clinic has about 7.5 pieces of digital equipment, there is significant variation, indicating differing levels of digital adoption across clinics. Only three clinics use the full range of digital equipment (20 devices/equipment); however, six clinics do not use digital equipment at all. The median value of 7 pieces of equipment aligns closely with the mean, suggesting a relatively balanced distribution of digital technologies among the surveyed clinics.

### 3.3. Technology Readiness

Analogous to the moderate utilization of technical equipment within the dental clinics, the overall technology readiness of the participants also falls within a moderate range. This section presents the results on technology readiness as measured by the validated Technology Readiness Index (TRI 2.0) developed and licensed by Parasuraman and Colby [16,17].

The TRI 2.0 assesses the readiness of individuals to adopt and use new technologies and comprises the following five assessment areas [17]:*I.* *TRI-Score:* An overall measure of technology readiness.*II.* *Optimism:* The positive attitude towards technology and the belief that it offers more flexibility, efficiency, and control.*III.* *Innovativeness:* The tendency to be open to new technology and to implement it at an early stage.*IV.* *Discomfort:* An uncomfortable feeling caused by technology. This is caused by the impression of not having enough control over the technology or being overwhelmed by the complexity.*V.* *Insecurity:* Uncertainty, which is caused by skepticism regarding the reliability and functionality of technology. [17]

The mean and median values for each of these dimensions are summarized below in Figure 4.

The results of the statistical analysis are as follows:*I.* The mean TRI-Score of 3.23 suggests a moderate level of overall technology readiness among the participants, indicating that, on average, dentists are neither highly resistant nor highly inclined towards adopting new technologies. The median TRI-Score of 3.22 corroborates this finding.*II.* Optimism: With a mean score of 3.22 (±) 0.94 and a median of 3, participants generally have a positive attitude towards technology and believe it improves their efficiency and control.*III.* Innovativeness: The mean score of 2.97 (±) 1.09 and median of 3 indicate a moderate tendency among dentists to be technology pioneers.*IV.* Discomfort: A mean score of 2.55 (±) 0.95 and a median of 2 reflect a moderate level of discomfort with technology, suggesting some participants feel overwhelmed by it.*V.* Insecurity: The mean score of 2.81 (±) 1.07 and median of 3 show a moderate level of insecurity, indicating some skepticism about the reliability of technology.

Overall, the slightly higher scores in optimism and innovativeness suggest a generally positive attitude towards digitalization and technology readiness among the surveyed dentists. This positive outlook indicates a favorable environment for the adoption and integration of new technologies in dental clinics in Germany.

### 3.4. Correlation Between TRI-Score and Other Parameters

Building on the previous analysis, we also investigated the relationship between the TRI score and various parameters:TRI-Score vs. Age:
A very significant negative correlation (*p* < 0.001) was found between the TRI score and the age of the dentists. Younger dentists exhibited higher technology readiness, suggesting that younger professionals are more inclined towards embracing digital technologies.
TRI-Score vs. Clinic Type:
The correlation between the TRI score and the type of clinic was non-significant (*p* = 0.198), indicating that the clinic structure does not significantly influence technology readiness.
TRI-Score vs. Number of Employees:
There was a very significant positive correlation (*p* < 0.001) between the TRI score and the number of employees in the clinic. Dentists in clinics with larger teams exhibited higher technology readiness, indicating that bigger teams may have a greater capacity to support and integrate new technologies.
TRI-Score vs. Clinic Location:
The correlation between the TRI score and the location of the clinic (urban, suburban, rural) was not significant (*p* = 0.331), suggesting that location does not substantially affect technology readiness.
TRI-Score vs. Gender:
The correlation between the TRI score and the gender of the dentist was non-significant (*p* = 0.306), indicating no gender-based differences in technology readiness.
TRI-Score vs. Professional Development:
The correlation between the TRI score and the professional development of the dentists was found to be non-significant (*p* = 0.127), suggesting that continuing education alone may not significantly influence overall technology readiness.
TRI-Score vs. Number of Equipment:
A very significant correlation (*p* < 0.001) was observed between the number of equipment and the TRI score. Dentists in clinics with more equipment had higher technology readiness scores, underscoring the link between digital investment and overall technology readiness.

Table 2 summarizes the findings and correlations between the TRI-Score and the parameters.

### 3.5. Multivariate Analysis TRI-Score

Multiple linear regression with a 95% confidence interval was performed to evaluate the independent associations between demographic and organizational factors (clinic location) and technology readiness.

The model was statistically significant (F(3,196) = 14.10, *p* < 0.001), explaining 17.8% of the variance in TRI scores (R^2^ = 0.178).

Younger dentists exhibited significantly higher technology readiness (β = −0.0185, *p* < 0.001), male dentists had significantly higher TRI scores compared to female dentists (β = 0.1710, *p* = 0.022), and clinics located in urban areas showed higher technology readiness scores than those in rural areas (β = 0.0766, *p* = 0.030).

### 3.6. Correlation Between Number of Equipment and Other Parameters

This study also explored the correlation between the number of digital equipment in dental clinics and various parameters, including age, clinic type, number of employees, clinic location, gender, and professional development of the dentists. Table 3 summarizes these correlations.

Number of Equipment vs. Age:

The correlation between the number of equipment and the age of the dentists was found to be non-significant (*p* = 0.338), indicating that age does not significantly influence the extent of digital equipment utilization in clinics.

Number of Equipment vs. Clinic Type:

A highly significant correlation (*p* < 0.001) was observed between the number of equipment and the type of clinic. Multi-dentist clinics (MVZ) and group clinics had significantly more equipment compared to solo clinics, suggesting that larger clinic structures tend to invest more in digital technologies. Table 4 shows the distribution of digital equipment by clinic type, with the average difference between solo practices and multi-dentist clinics (MVZ) being approximately six devices/services.

Number of Equipment vs. Number of Employees:

There was a very significant positive correlation (*p* < 0.001) between the number of equipment and the number of employees in the clinic. This finding indicates that clinics with more staff are likely to have a greater number of digital equipment, possibly to support the higher operational demands.

Number of Equipment vs. Clinic Location:

The correlation between the number of equipment and the location of the clinic (urban, suburban, rural) was not significant (*p* = 0.152), suggesting that geographical location does not make a major contribution to the adoption of digital technologies.

Number of Equipment vs. Gender:

The correlation between the number of equipment and the gender of the dentist was found to be non-significant (*p* = 0.278), indicating no gender-based differences in the adoption of digital technologies.

Number of Equipment vs. Professional Development:

A significant (*p* = 0.022) but weak correlation was found between the number of equipment and the professional development of the dentists. Dentists who participated in more continuing education tended to have slightly more equipment, suggesting that ongoing education may play a role in supporting digital adoption.

### 3.7. Digitalization Types

In this study, clinics were categorized based on the number of digital devices used, ranging from 1 to 20. To better understand the differences in digital adoption, the clinics were divided into four distinct digitalization types, corresponding to the quartiles of device usage (see Figure 5). This classification provided a clear structure for analyzing how digital integration varies across dental clinics.

1Low Adopters (1st Quartile) (n = 44):

Clinics in this group are characterized by minimal digitalization, using a median of just two devices. These clinics primarily consist of solo practices, with no MVZs represented in this category. The Technology Readiness Index (TRI) score for this group is the lowest among all types, with a score of 2.8, reflecting limited enthusiasm and preparedness for adopting new technologies.

2Moderate Adopters (2nd Quartile) (n = 63):

Clinics in the second quartile show a moderate level of digitalization, with around six devices on average. Although there is an increase in device usage compared to the first group, these clinics still show relatively cautious adoption of technology. The TRI-Score in this group is slightly higher, scoring 3.26 on average, indicating a gradual shift in readiness towards embracing more digital tools.

3Advanced Adopters (3rd Quartile) (n = 43):

Representing clinics with 8–10 devices on average, the third quartile demonstrates a more proactive approach to digitalization. These advanced adopters display slightly higher TRI-Scores (3.29), reflecting a stronger inclination toward integrating new technologies. While solo practices still appear in this group, there is an increasing presence of larger clinics.

4Power Users (4th Quartile) (n = 50):

Clinics in the top quartile, the Power Users, utilize the most digital devices, with some reaching up to 20. This group exhibits the highest TRI-Scores, with a mean of 3.51, underlining a clear readiness to adopt and integrate a wide range of digital tools. Interestingly, MVZs are disproportionately represented in this quartile, making up twice as many clinics as in other quartiles, while solo practices are almost absent. This correlation is very significant (*p* < 0.001). Furthermore, this group includes a significantly higher proportion of male dentists (13 women, 37 men). The significance of the correlation amounted to 0.024. No significant differences or correlations were found between the groups (quartiles) with respect to age.

The clear upward trend in TRI-Scores across the quartiles demonstrates that as clinics adopt more digital devices, their overall technology readiness increases correspondingly. This progression from Low Adopters to Power Users suggests that technology readiness is both a driver and a consequence of more extensive digital integration.

Building on these findings, distinct patterns of technology readiness and digital adoption emerge within German dental clinics. Clinic size, professional training, and financial resources play a significant role in shaping digital integration. While foundational digital tools such as digital radiography and CAD/CAM systems are widely implemented, the adoption of advanced technologies, including AI-based diagnostics and 3D printing, remains limited. These insights raise important questions about the factors influencing digital transformation and the barriers preventing its widespread adoption, which will be explored in the following discussion.

## 4. Discussion

This research gives valuable insights into the current state of digital technology adoption and technology readiness among German dentists, contributing to the broader discourse on digital transformation in healthcare. Building on previous research, which identified five major technologies shaping the digital transformation of dentistry, this paper focuses on technology readiness as a critical driver of digital adoption [15]. Together, these studies establish a comprehensive framework for understanding the ongoing digital transformation in dental practices, emphasizing both current adoption patterns and future implications. At the same time, the increasing integration of digital technologies into healthcare systems is impacting the wider population, for example, by improving access to services, enabling patients to participate more actively in their care, and creating new expectations for digital treatment [1,46,47].

The first phase of this research revealed that the relevance of digital trends, both in terms of clinical applications and economic impact, is steadily increasing and is projected to play a pivotal role by 2030 [15]. Despite the increased and expected importance, there is still very limited high-quality research on the top trends, such as AI [14]. This matches the moderate overall technology readiness observed, with higher scores among younger dentists and those in larger, multi-dentist clinics, and reflects established trends in the healthcare sector, where younger professionals exhibit a greater openness to adopting new technologies [48,49]. Recent studies also suggest that the integration of digital elements into dental education significantly shapes young dentists’ interest in, and perceived competence with, new technologies, further reinforcing their greater technology readiness [50]. Additionally, the significant correlation between clinic size and technology readiness aligns with findings from general management research, which demonstrate that larger organizations, including SMEs, are more likely to adopt digital tools due to economies of scale and access to shared resources [9,51,52]. However, the findings also indicate that significant efforts will be required to ensure these advancements become a reality in clinical practice by 2030, particularly in addressing disparities among smaller clinics and enhancing professional training.

A particularly notable finding is the strong association between clinic size and digital device adoption. Larger clinics, such as multi-dentist clinics (MVZ) and group clinics, displayed significantly higher levels of technology readiness and digital equipment usage. This suggests that the financial and operational advantages inherent in larger clinics—such as pooled resources, collaborative infrastructure, and economies of scale [53]—could enable greater investment in digital technologies. These patterns echo similar dynamics observed in hospitals, where size and complexity often correlate with higher adoption rates of digital innovations [54,55,56]. This association underscores the importance of targeted strategies to bridge the digital divide, ensuring that smaller and solo clinics are not left behind in the digital transformation of dentistry.

Conversely, the study identified non-significant correlations between technology readiness and both geographical location and gender in our bivariate correlation analysis, indicating that these are not major factors in the introduction of digital technology in dental clinics. This finding contrasts with previous studies that have reported gender-specific differences in digital adoption [21,57,58]. However, when applying multivariate analysis, controlling for age and other variables, both gender and the clinic’s classification as urban or rural emerged as significant independent predictors of technology readiness. This underlines the importance of multivariate modeling in revealing hidden associations that may not be detectable through simple bivariate correlations and highlights the necessity of adjusting for confounding factors when analyzing determinants of digital adoption [59].

In addition, the weak but significant correlation between professional development and digital equipment usage suggests that while continuous education is beneficial, it may need to be more precisely targeted toward digital competencies to drive substantial adoption [60,61]. This highlights an opportunity for education providers and professional organizations to refine their training programs to address specific digital skills and technologies.

The positive attitude towards digitalization among younger dentists and those in larger clinics is encouraging and indicates a promising trajectory for the future of dental technology. However, this trend also exposes disparities in access to and utilization of digital tools, with smaller or solo clinics being less equipped and prepared for digital adoption. Bridging this gap will be essential for ensuring equitable access to advanced dental care. Policies aimed at financial support, such as subsidies for digital equipment, and the development of training programs tailored to smaller clinics could help address these disparities. Furthermore, fostering collaborative networks and resource-sharing initiatives may empower smaller clinics to adopt digital technologies more effectively [62,63].

### Limitations and Future Directions

While this study provides a comprehensive analysis of digital adoption and technology readiness, several limitations must be acknowledged. The cross-sectional design limits the ability to establish causal relationships between technology readiness and digital adoption [64]. Longitudinal studies are recommended to support the cross-sectional design in order to investigate how these factors evolve over time and to examine the long-term impact of digitalization on dental clinic efficiency, patient outcomes, and overall healthcare delivery [64].

Another limitation lies in the geographic concentration of this study in Germany. While the findings provide valuable insights into the German dental sector, the results may not be generalizable to other countries or regions with different healthcare systems and levels of digital adoption. Future research should extend these analyses to diverse international contexts to validate the findings and identify region-specific trends and challenges.

Furthermore, other confounding factors might potentially be relevant. These include years of professional experience, income level, and the availability of financial resources for technology investments, all of which could influence technology readiness. Future studies should aim to include a broader range of individual and organizational factors to gain an even more detailed understanding of digital adoption in dental clinics.

Finally, the study highlights the need to explore the barriers faced by clinics with lower technology readiness. Further research could focus on identifying the specific challenges that smaller and less resourced clinics encounter and test targeted interventions to overcome these obstacles. Understanding these barriers will be critical for ensuring that the benefits of digital transformation are accessible to dental clinics, regardless of size, ownership, or location.

## 5. Conclusions

This study provides a comprehensive analysis of technology readiness and digital adoption in German dental clinics, identifying key factors that influence the integration of digital tools in daily practice. By applying the Technology Readiness Index (TRI 2.0), the study examined how age, clinic type, and professional development impact the adoption of digital equipment, revealing clear disparities in technology readiness among dental professionals.

The results show that larger multi-dentist clinics (MVZs) demonstrate significantly higher levels of digital technology adoption compared to solo practices. Among the most widely used digital technologies, digital radiography, electronic patient records, and CAD/CAM systems were identified as the most prevalent, suggesting that foundational digital infrastructure is already well established in many clinics. However, advanced technologies such as AI-assisted diagnostics, 3D printing, and fully digital workflows remain underutilized, with adoption rates significantly lower than expected. This indicates that while the industry has made progress in integrating digital solutions, the adoption of more complex and emerging technologies is still in its early stages.

Several key factors drive digital adoption in dental clinics. Dentists who score higher in TRI 2.0 subcategories such as optimism and innovativeness are more likely to integrate digital tools into their workflows. Additionally, younger professionals and those engaged in ongoing digital training programs exhibit higher technology readiness and a greater willingness to implement new solutions. Clinic size also plays a crucial role, as larger clinics often have more resources to invest in new technologies and infrastructure.

Despite these positive trends, several barriers to digital transformation persist. Financial constraints remain a major challenge, particularly for smaller clinics, which could be due to the struggle with the high upfront costs of digital equipment. Furthermore, limited training opportunities and a lack of structured education programs contribute to hesitation in adopting newer digital solutions, as many clinicians feel inadequately prepared to integrate them into their workflows.

## Figures and Tables

**Figure 1 healthcare-13-01155-f001:**
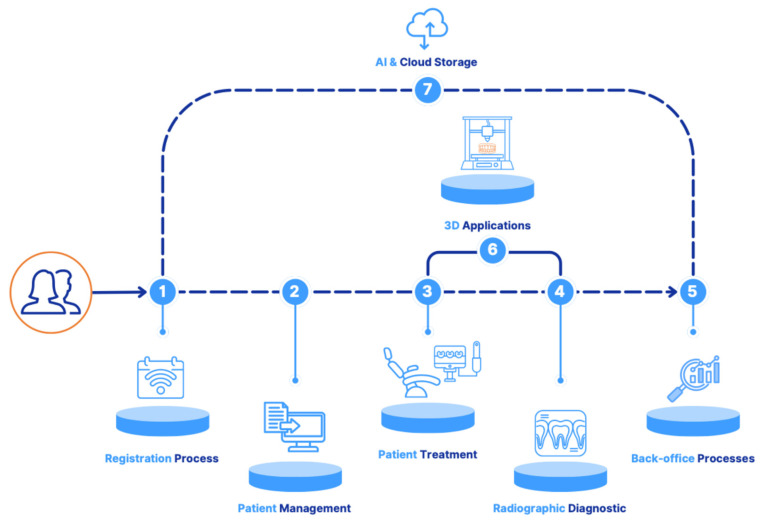
Patient journey with digital equipment visualizing the seven sections of the digital journey.

**Figure 2 healthcare-13-01155-f002:**
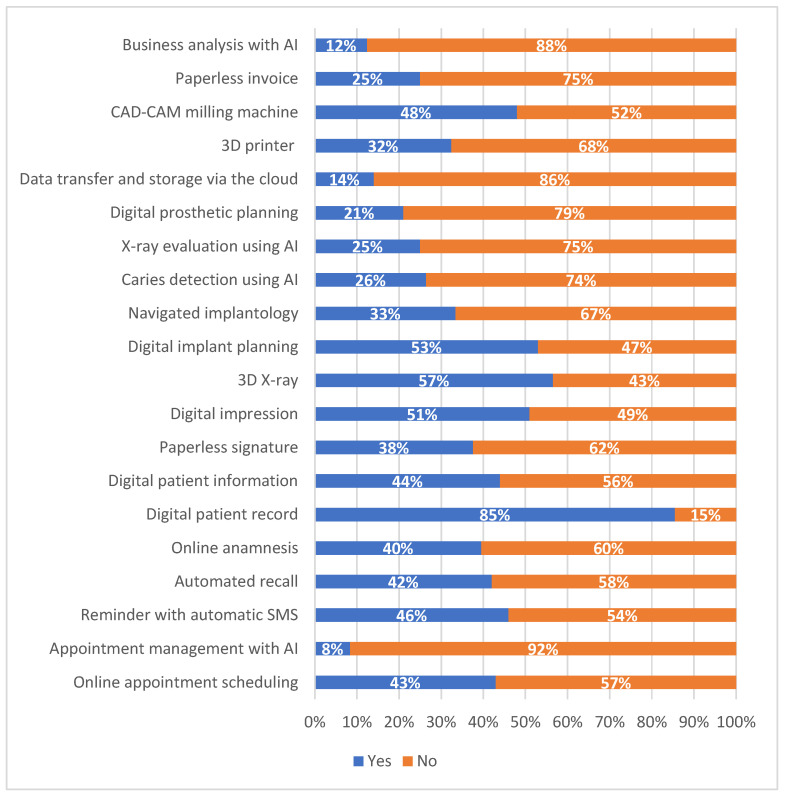
Usage of digital equipment (devices/services) in dentistry.

**Figure 3 healthcare-13-01155-f003:**
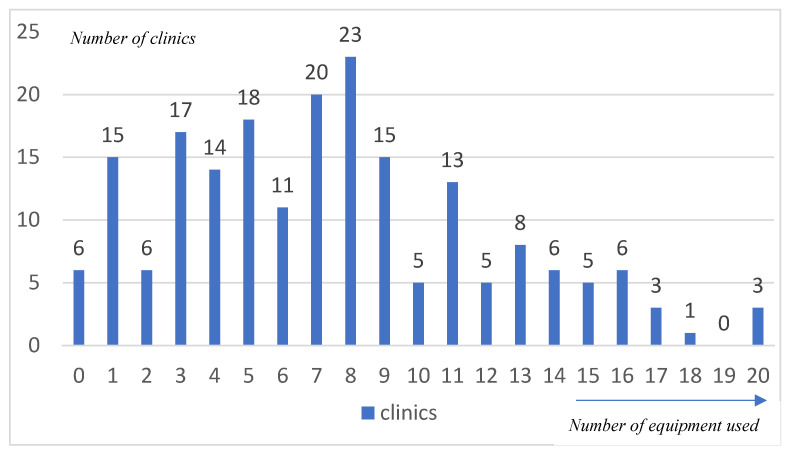
Distribution of digital equipment per clinic.

**Figure 4 healthcare-13-01155-f004:**
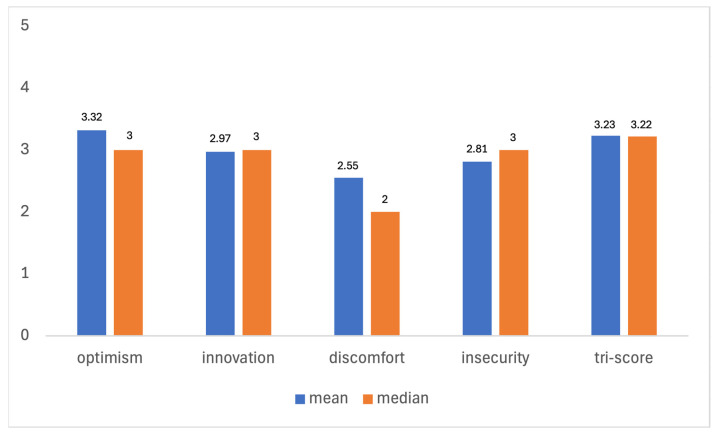
Technology Readiness Index (TRI 2.0) Scores.

**Figure 5 healthcare-13-01155-f005:**
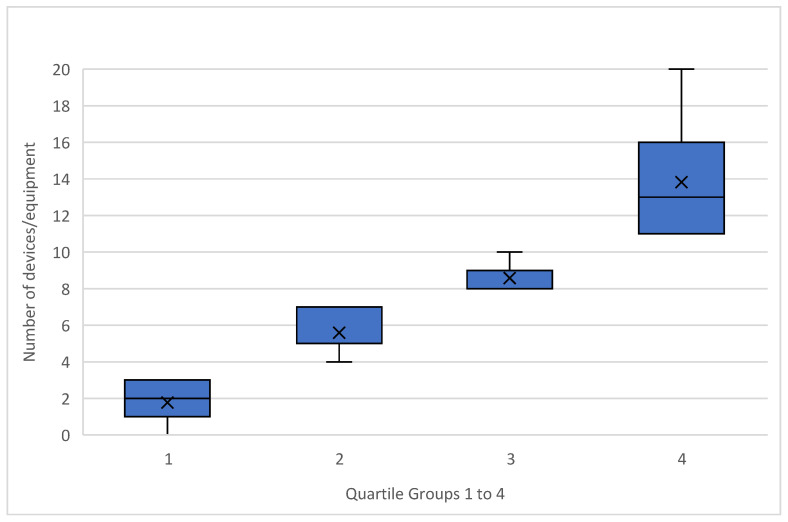
Distribution of digital equipment usage across quartile groups: median device counts by clinic type.

**Table 1 healthcare-13-01155-t001:** The age distribution is shown.

Gender	Age (Mean)	Age (Median)	SD
female	37.7	35	(±) 10.18
male	45.4	46	(±) 12.97

**Table 2 healthcare-13-01155-t002:** Correlation analysis of the TRI-Score vs. other parameters.

*Correlation–Analysis*	Age	Clinic Type	Number of Employees	Clinic Location	Gender	Professional Development	Number of Equipment
TRI-Score	**−0.338**^i^ (*p* < 0.001)	123.406 (111) ^ii^ (*p* = 0.198)	**0.316**^i^ (*p* < 0.001)	116.935 (111) ^ii^ (*p* = 0.331)	40.832 (37) ^ii^ (*p* = 0.306)	167.785 (127) ^ii^ (*p* = 0.127)	**0.384**^i^ (*p* < 0.001)

Legend: ^i^: Spearman-Rho correlation coefficient; ^ii^: chi-square test.

**Table 3 healthcare-13-01155-t003:** Correlation analysis number of digital equipment vs. other parameters.

*Correlation–Analysis*	Age	Clinic Type	Number of Employees	Clinic Location	Gender	Professional Development
Number of Equipment	−0.068 ^i^ (*p* = 0.338)	**121.391 (57)**^ii^ (*p* < 0.001)	**0.388**^i^ (*p* < 0.001)	67.938 (57) ^ii^ (*p* = 0.152)	22.130 (57) ^ii^ (*p* = 0.278)	**102.780 (76)**^ii^ (*p* = 0.022)

Legend: ^i^: Spearman-Rho correlation coefficient; ^ii^: chi-square test.

**Table 4 healthcare-13-01155-t004:** Average number of digital equipment by clinic type.

Clinic Type	Average Number of Equipment/Medians
Solo clinic	4.96/3
Group clinic	7.27/7
Multi-dentist (MVZ)	11.2/11
University clinic	8.03/8

## Data Availability

The data supporting the findings of this study are available from the authors upon reasonable request. Access will be granted in cases of legitimate interest, and the data remain the property of the authors.

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
