# Peer review of "Technology Readiness Drives Digital Adoption in Dentistry: Insights from a Cross-Sectional Study"

_healthcare, 2025, doi:10.3390/healthcare13101155_

Round 1
Reviewer 1 Report
Comments and Suggestions for Authors
Reviewer comments healthcare-3567079
General:
In this study, the authors have basically compiled data on a current and relevant topic using a validated method. Important observations have been made which may have implications on future clinical practice within the scope of the explored location/ cohort and at a wider level.
Abstract:
- Seems to the point and is well written.
Introduction/ Background:
- The use of BOLD letters to emphasize specific words and statements should be avoided in the manuscript text, except for key words with bullet points (example. Optimism, innovativeness…etc)
Materials & Methods:
- The dentist population was from the entire country or a specific region?
Does Germany mean the whole country? How were the dentist numbers distributed across the 16 different states, if they were all included?
- The online form used for collecting the information on all three parameters should be included in the appendix.
Reviewer 2 Report
Comments and Suggestions for Authors
Dear authors,
I evaluated the article titled “Technology Readiness Drives Digital Adoption in Dentistry: Insights from a Cross-Sectional Study”. The goal was "to assess the relationship between technology readiness and digital technology adoption among German dentists, focusing on the impact of clinic characteristics and professional development”.
- adjust the font size throughout the text, please.
- it is not necessary bold letters in the middle of the text. Orr do you have any justification for it?
REFERENCES
- 46% of the refs were published in the last 5 years. This topic can be considered relatively new… then, I suggest to increase the %number of updated articles.
INTRO: this section is interesting ans well-developed; suggestions of recent articles in the area for update
- A recent article could be used for the intro and discussion
(ref suggested) Technology Readiness Level of Robotic Technology and Artificial Intelligence in Dentistry: A Comprehensive Review. Surgeries 2024, 5, 273–287. https://doi.org/10.3390/surgeries5020025
- for the second paragraph, another new study is recommended below
(new ref. suggested) The Impact of Technology Teaching in the Dental Predoctoral Curriculum on Students’ Perception of Digital Dentistry. Dent. J., 2024, 12, 75. https://doi.org/10.3390/dj12030075
- remove bullet points and describe them in the text.
M&M
- please clarify the reliability of this tool "using the tool online tool LimeSurvey"
- was the survey previously validated?
RESULTS
- in Tables 1 and 4, and Figure 4, change comma per point (numbers)
DISCUSSION: it could be expanded
- 1st paragraph… showing the impact of digital technologies in this era on the population.
CONCLUSION: I considered this part longer than normal… please shorten it.
Any papers recommended in the report are for reference only. They are not
mandatory. You may cite and reference other papers related to this topic.
Reviewer 3 Report
Comments and Suggestions for Authors
Dear Authors,
The authors focused on an earlier study, reference 13, and it should be detailed. It seems to be the work of the same research team. Is it?
The technological readiness index (TRI) should be related to its practical application, and the authors should present the settings related to the TRI.
The rationality of the aim of the present study is not clear. Did the authors intend to focus on German dentists? did the authors select a private dental practice?
I propose the articles: 1)Digital solutions for migrant and refugee health: a framework for analysis and action doi: 10.1016/j.lanepe.2024.101190, and 2) Digital intraoral and radiologic records in forensic identification: Match with disruptive technology doi: 10.1016/j.forsciint.2024.112104
In the materials and methods section, the authors did not clarify the technological devices or technology used. Who made the simple selection? Inclusion and exclusion criteria? Professional qualifications and habilitation? Etc...
Figure 1 is not clear and we do not understand its inclusion in the methodological process.
The results section is not correct. The authors should present the results for future discussion in the discussion section. Figure 2 is the relevant one! Figure 3 is not clear!
The discussion section should be improved and similar articles should be discussed. The authors' highlights are intuitive, it is expected that a general analysis of the literature will discuss its rationality and focus on issues related to specific practice and emerging technologies management and evaluation.
The conclusion should be brief and objective.
The font and type used should be carefully analysed.
Round 2
Reviewer 2 Report
Comments and Suggestions for Authors
Dear authors,
Thank you for your responses and adjustments.
Author Response
Dear reviewer,
Thank you very much for the very good comments on optimising the article.